# Frustrated Lewis pair catalyst realizes efficient green diesel production

De-Chang Li[1,2,3,4], Zhengyi Pan[1,2,3,4], Zhengbin Tian[1,2,3], Qian Zhang[1,2,3,4], Xiaohui Deng[1,2,3], Heqing Jiang[1,2,3,4] & Guang-Hui Wang [1,2,3,4] ✉

Hydrotreating renewable oils over sulfided metal catalysts is commercially applied to produce green diesel, but it requires a continuous sulfur replenishment to maintain catalyst activity, which inevitably results in sulfur contamination and increases production costs. We report a robust P-doped NiAl-oxide catalyst with frustrated Lewis pairs (i.e., P atom bonded with the O atom acts as an electron donor, while the spatially separated Ni atom acts as an electron acceptor) that allows efficient green diesel production without sulfur replenishment. The catalyst runs more than 500 h at a weight hourly space velocity (WHSV) of 28.3 h$^{-1}$ without deactivation (methyl laurate as a model compound), and is able to completely convert a real feedstock of soybean oil to diesel-range hydrocarbons with selectivity >90% during 500 h of operation. This work is expected to open up a new avenue for designing non-sulfur catalysts that can make the green diesel production greener.

Green diesel, also called renewable diesel, is a mixture of oxygen-free hydrocarbons produced from renewable oils (e.g., vegetable oils and animal fats)[1,2], which is chemically identical to petroleum diesel and fully compatible with the existing engines[3]. Using green diesel instead of petroleum diesel can significantly reduce the $CO_2$ emission by over 50%[4–6], contributing to mitigation of climate change as well as energy scarcity[7]. Currently, the global production capacity of green diesel, mainly by hydrotreating technology (commercialized by Neste Oil, UOP/ENI, Haldor-Topsoe, etc.)[8], is more than 2.61 billion gallons per year and will reach 7.45 billion gallons in 2027[9]. The commercial hydrotreating processes for green diesel production often operate at temperatures of 260–400 °C under $H_2$ pressures of 3–10 MPa using sulfided metals (e.g., sulfided CoMo and NiMo) as catalysts[10]. During operation a continuous replenishment of sulfur source (such as $H_2S$) is required to maintain the activity of sulfided catalysts, which undesirably results in sulfur contamination and increases production costs[11].

To address this issue, sulfur-free catalysts[12,13], including noble metals (e.g., supported Pd, Pt catalysts)[14,15], transition metal carbides/phosphides[16,17], etc., have been investigated in the past decades. Although noble metal catalysts show high activity[18], the coke-induced rapid deactivation and their high costs represent the major obstacles to the industrial implementation[19,20]. Relative to noble metal catalysts,

transition metal phosphides afford comparable activity and relatively good stability[21], and thus are considered as promising catalysts for hydrotreating processes[22]. Taking $Ni_2P$-based catalysts as examples, their activity relies on the existence of metallic Ni centers and acidic properties (Brönsted acidity caused by P-OH groups and/or Lewis acidity due to electron transfer from Ni to P)[23], between which the synergistic effect is beneficial for hydrotreating reactions[24–28]. Nevertheless, these phosphided catalysts also suffer from deactivation, due to carbon deposition, oxidation by $H_2O$ or oxygen-containing compounds, sintering and phosphide phase changing, etc.[12,25], which are unable to meet the industrial requirements. Thus, new types of sulfur-free catalysts, taking into account the long-term stability and catalytic efficiency as well as the reproducibility and scalability of the synthetic methods, should be designed for green diesel production.

In this study, we report a monolithic P-doped NiAl-oxide catalyst with a hierarchical porous structure (derived from the *maize straw (ms)* template, Fig. 1a and Supplementary Fig. 1) for green diesel production. We first prepared NiAl layered double hydroxide nanosheets anchored on the *ms* cell walls (*ms*-NiAl-LDHs, Supplementary Fig. 2) by hydrothermal process. Then, we calcinated the obtained *ms*-NiAl-LDHs at 800 °C under air to remove the *ms*-template, and to obtain the NiAl-mixed oxide with a biological

[1]Qingdao Institute of Bioenergy and Bioprocess Technology, Chinese Academy of Sciences, Qingdao, China. [2]Shandong Energy Institute, Qingdao, China. [3]Qingdao New Energy Shandong Laboratory, Qingdao, China. [4]University of Chinese Academy of Sciences, Beijing, China. ✉e-mail: wanggh@qibebt.ac.cn

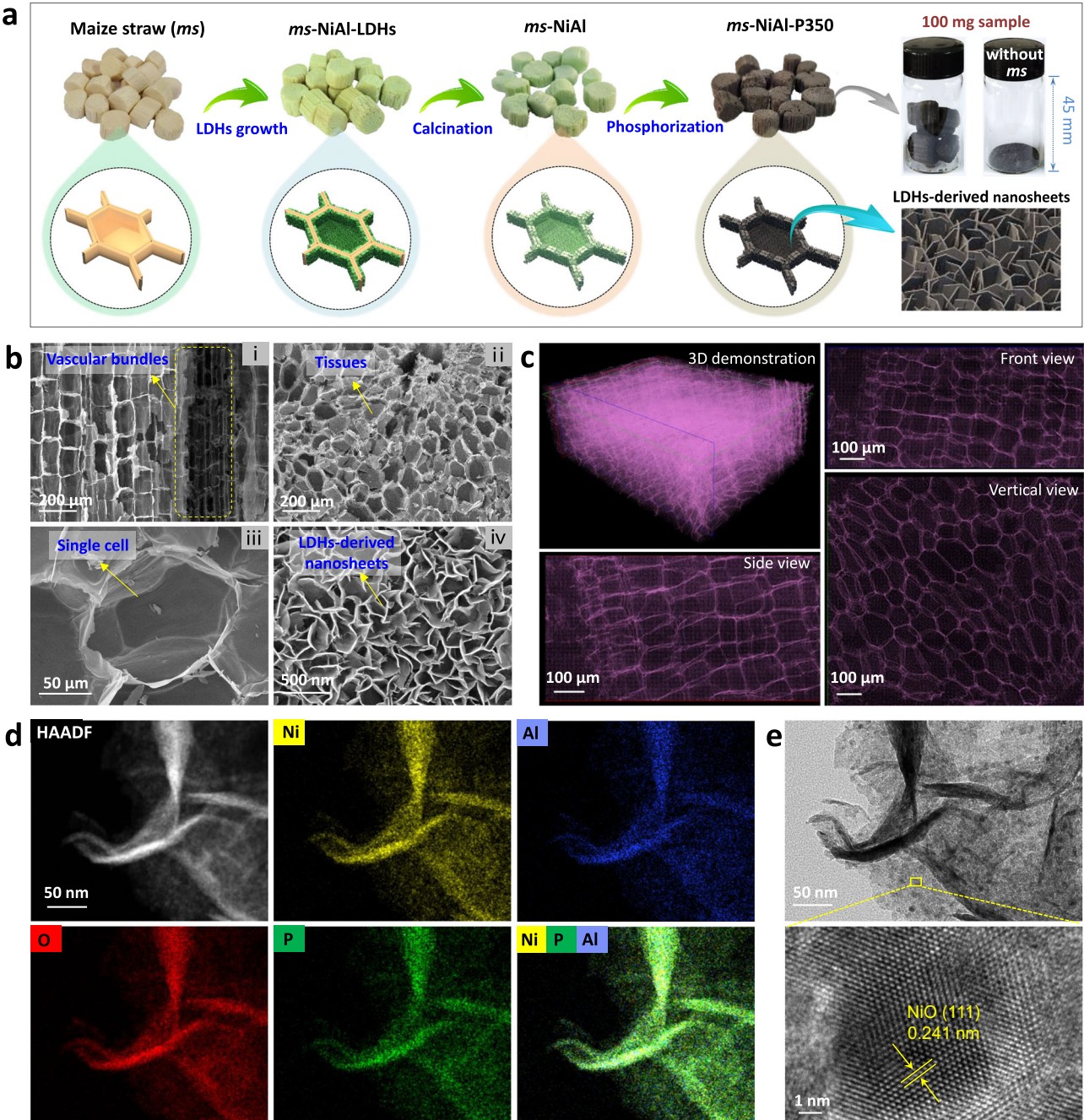

**Fig. 1 | Synthesis and structural characterization of *ms*-NiAl-P350. a** Schematic illustration of the synthetic procedure. Photographs show the samples of *ms*-NiAl-P350 monolith and NiAl-P350 powder (synthesized without *ms*-template) with the same mass, implying the relatively low apparent density of *ms*-NiAl-P350. **b** SEM and (**c**) synchrotron X-ray tomographic images of *ms*-NiAl-P350, showing the biological architecture replicated from *ms*-template and the nanosheet structure derived from LDHs. **d** HAADF-STEM and EDS elemental mapping images of *ms*-NiAl-P350, showing that the Ni, Al, O, and P elements are highly mixed with one another. **e** TEM (up) and HRTEM (down) images of *ms*-NiAl-P350, the lattice spacing fits well to the (111) plane of NiO.

architecture (*ms*-NiAl, Supplementary Figs. 3 and 4). The calcination step, i.e., fabrication of the mixed NiAl-oxide structure, is critical upon phosphorizing *ms*-NiAl (NaH$_2$PO$_2$ as P source, 350 °C), whereby P-doped NiAl-oxide catalyst (*ms*-NiAl-P350) rather than nickel phosphide is obtained. The *ms*-NiAl-P350 catalyst runs more than 500 h using methyl laurate as feedstock (WHSV = 28.3 h$^{-1}$) without deactivation being observed (85–88% conversion with selectivity towards hydrocarbons >96% during 500 h of operation). The catalyst can be further extended to convert the real feedstocks, e.g., soybean oil, palm oil, duck fat, and waste cooking oil. We presume that the benign interaction between P and NiAl-oxide

(i.e., frustrated Lewis pair of Ni(-O-Al)···P) is a critical factor in achieving coke- and sintering-resistant activity during hydrotreating reactions.

## Results

### Synthesis and characterizations of catalysts

The geometric shape of *ms*-NiAl-P350 was unchanged after phosphorizing (Supplementary Figs. 5 and 6), but the color changed from light green to black (Fig. 1a), implying the successful incorporation of P (Supplementary Tables 1 and 2). Scanning electron microscopy (SEM) and synchrotron X-ray tomography images (Fig. 1b, c) also revealed

that the *ms*-NiAl-P350 displayed a biological architecture perfectly replicated from the *ms*-template, where the naturally well-developed mass transfer channels (e.g., honeycomb-like tissues and vascular bundles) with diameters ranging of 100–200 μm (hardly being fabricated by conventional chemical means) were well retained. The Brunauer-Emmett-Teller (BET) surface area of *ms*-NiAl-P350 is 103.4 m²·g⁻¹, which is lager than that of NiAl-P350 powder (82.4 m²·g⁻¹, synthesized without *ms*-template, Supplementary Fig. 7). Due to the high porosity, the *ms*-NiAl-P350 exhibited a much lower apparent density of 11.7 mg·cm⁻³ (Fig. 1a, Supplementary Fig. 8 and Supplementary Table 3) than the 307.1 mg·cm⁻³ of NiAl-P350 powder. High angle annular dark-field scanning transmission electron microscopy (HAADF-STEM) and the corresponding elemental mapping images (Fig. 1d) revealed that the Ni, Al, O, and P elements in *ms*-NiAl-P350 were highly mixed with one another. It is expected that such uniform elemental distributions were very conducive to the fabrication of active centers involving Ni and P. Transmission electron microscopy (TEM) and high-resolution TEM (HRTEM) images (Fig. 1e) displayed that the lattice spacing of the nanocrystals was 0.241 nm, corresponding to the (111) plane of NiO, rather than the nickel phosphide phase.

X-ray diffraction (XRD) was performed to investigate the crystalline structure of *ms*-NiAl-P350. Its XRD pattern only showed the peaks belonging to the NiO phase (Fig. 2a), without the diffraction peaks ascribed to the Al- and/or P-related phases. For comparison, phosphorization of NiO supported on Al₂O₃ under the same condition led to the transformation of NiO into nickel phosphide phase (Supplementary Fig. 9a), being in line with previous literature[29,30]. This difference was attributable to the formation of Al³⁺ doped NiO-type phase of *ms*-NiAl[31], wherein the interaction between NiO and Al³⁺ stabilizes the NiO-type phase against phosphorization (also against reduction by H₂, Supplementary Fig. 9b). Ni phosphide nanocrystals were formed by increasing the phosphorization temperature to 450 °C (*ms*-NiAl-P450, Fig. 2a and Supplementary Fig. 10), or by reducing the

*ms*-NiAl under H₂/Ar flow at 800 °C (to break the interaction between NiO and Al³⁺) followed by phosphorization at 350 °C (*ms*-NiAl-H₂800-P350, Supplementary Fig. 11).

X-ray photoelectron spectroscopy (XPS) and X-ray absorption fine structure (XAFS) characterizations were performed to investigate the chemical states of elements in *ms*-NiAl-P350 and the reference samples. The Ni 2p₃/₂ XPS spectrum of *ms*-NiAl (Fig. 2b and Supplementary Table 4) showed three peaks at 856.9 eV, 855.7 eV, and 854.3 eV, which were assigned to Ni³⁺, Niᵟ⁺ (2 < δ < 3, due to the formation of Ni−O−Al bonds, Supplementary Fig. 12) and Ni²⁺, respectively[31,32]. We found that the binding energies of the three peaks shifted to lower values gradually after phosphorizing at 300 °C (*ms*-NiAl-P300) and 350 °C (*ms*-NiAl-P350), due to the contribution of P-doping. A new peak at 852.5 eV was observed under a higher phosphorization temperature (450 °C, *ms*-NiAl-P450) and could be assigned to the Ni species in Ni₂P phase (Fig. 2b). In P 2p region, *ms*-NiAl-P350 showed a peak at 133.8 eV, which arises from P-O species; while a new peak (129.5 eV) belonging to the P species in Ni₂P was observed for *ms*-NiAl-P450 (Fig. 2c). Correspondingly, the binding energy of Al 2p for *ms*-NiAl-P450 shifted to higher value (belonging to Al³⁺ in AlPO₄, Fig. 2d) compared with that for *ms*-NiAl-P350, implying the loss of interaction between Al³⁺ and NiO. These results indicated that the P element in *ms*-NiAl-P350 modified the electronic structure of Ni species in NiAl-mixed oxide without forming the nickel phosphide phase.

The Ni K-edge XANES spectrum of *ms*-NiAl was very close to that of NiO (Fig. 2e), but the white line exhibited a higher intensity, implying a more oxidized Ni species (due to the formation of Ni−O−Al bonds, being in line with the XPS data, Fig. 2b). The white line intensity of *ms*-NiAl-P350 was between those of NiO and *ms*-NiAl (Fig. 2e), indicating a slight decrease of the Ni oxidation state after phosphorization. As expected, the XANES spectrum of *ms*-NiAl-450 was close to that of Ni foil, but the near-edge feature shifted to higher energy (Fig. 2e), implying the formation of Ni₂P phase (electron transfer from Ni to P).

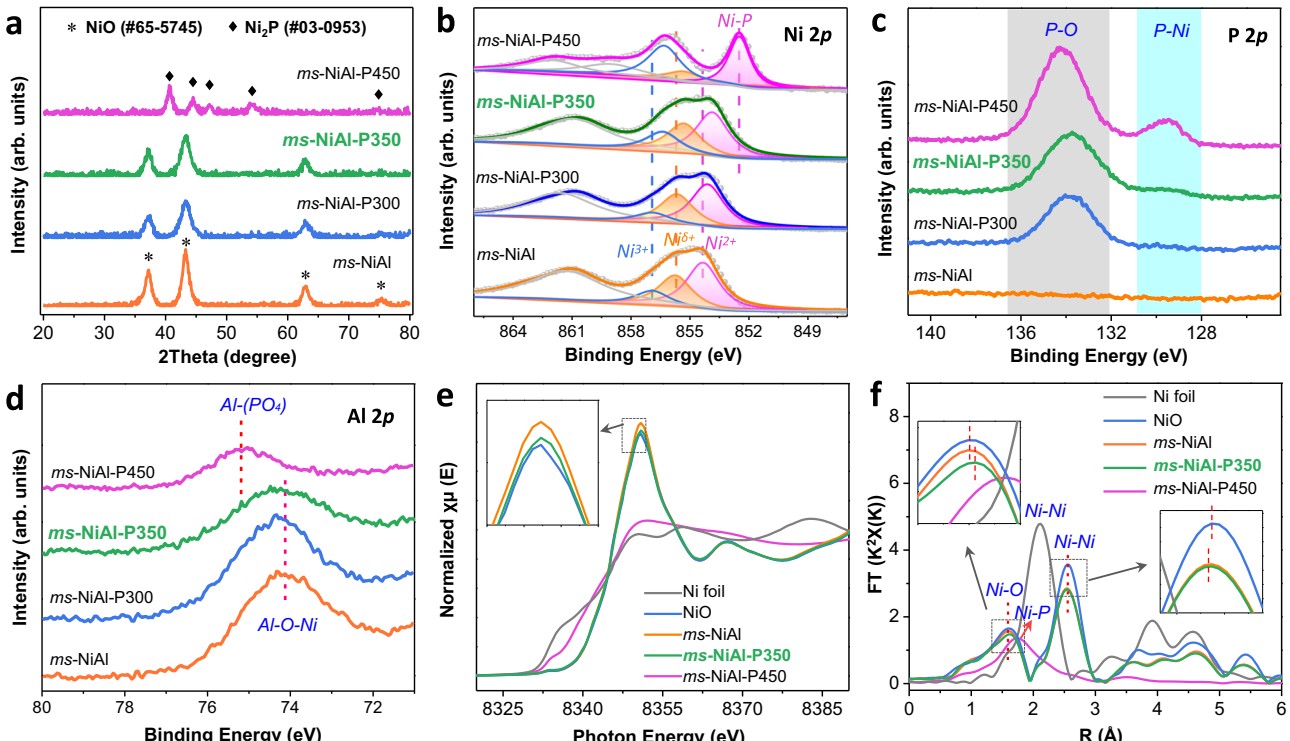

**Fig. 2 | Structural characterization of *ms*-NiAl-P350. a** XRD results of *ms*-NiAl-P350 and the reference samples. XPS spectra in (**b**) Ni 2p₃/₂ region, (**c**) P 2p region, and (**d**) Al 2p region of *ms*-NiAl-P350 and the reference samples. Ni K-edge (**e**) XANES and (**f**) EXAFS spectra of *ms*-NiAl-P350 and the reference samples. The inset of (**e**) shows the enlarged XANES white-line region. The insets of (**f**) show the enlarged Ni−O and Ni−Ni peak regions.

The Ni K-edge EXAFS curves (Fig. 2f and Supplementary Table 5) showed that *ms*-NiAl-P350 contained two peaks belonging to the Ni−O and Ni−Ni bonds in NiO, similar to those of *ms*-NiAl and NiO. However, the Ni−O peak shifted to higher R value (as opposed to the Ni−Ni peak), and this can be attributed to the P-doping. Again, these results evidenced the presence of P-doped NiAl-oxide structure in *ms*-NiAl-P350.

Based on these features, we constructed and optimized the model of P-doped NiAl-oxide using density functional theory (DFT) calculations to provide theoretical insights into the electronic structure of *ms*-NiAl-P350 (Fig. 3a and Supplementary Fig. 13). The Bader charge and charge density differences (Supplementary Figs. 14 and 15) showed that electron transfers from Ni−O to the nearby O−Al, leading to a more positive Ni site, and the P atom which preferentially bonds with the O atom acts as

electron donor (being consistent with the XPS and XAFS results). The resulting spatially separated Ni and P sites with a distance of 3.7 Å constitute a frustrated Lewis pair (FLP) (Fig. 3a and Supplementary Fig. 16). As revealed by DFT calculations, when a $H_2$ molecule approaches the constructed FLP site, electron transfer occurs along the path of P → H1 → H2 → Ni, resulting in the electron loss of P atom (0.17 eV) and the electron gain of H1 atom (0.09 eV), H2 atom (0.03 eV), and Ni atom (0.05 eV) (Fig. 3b and Supplementary Fig. 17), thereby leading to the elongation of the H−H bond from 0.78 Å in gaseous $H_2$ to 1.83 Å in $H_2$ (TS). Then, the $H_2$ molecule is heterolytically dissociated to form a stable H−Ni and H−P configuration (Fig. 3c). From the energy profile depicted by Fig. 3c, the FLP facilitates the heterolytic H−H scission with significantly lower activation barrier (0.32 eV) than

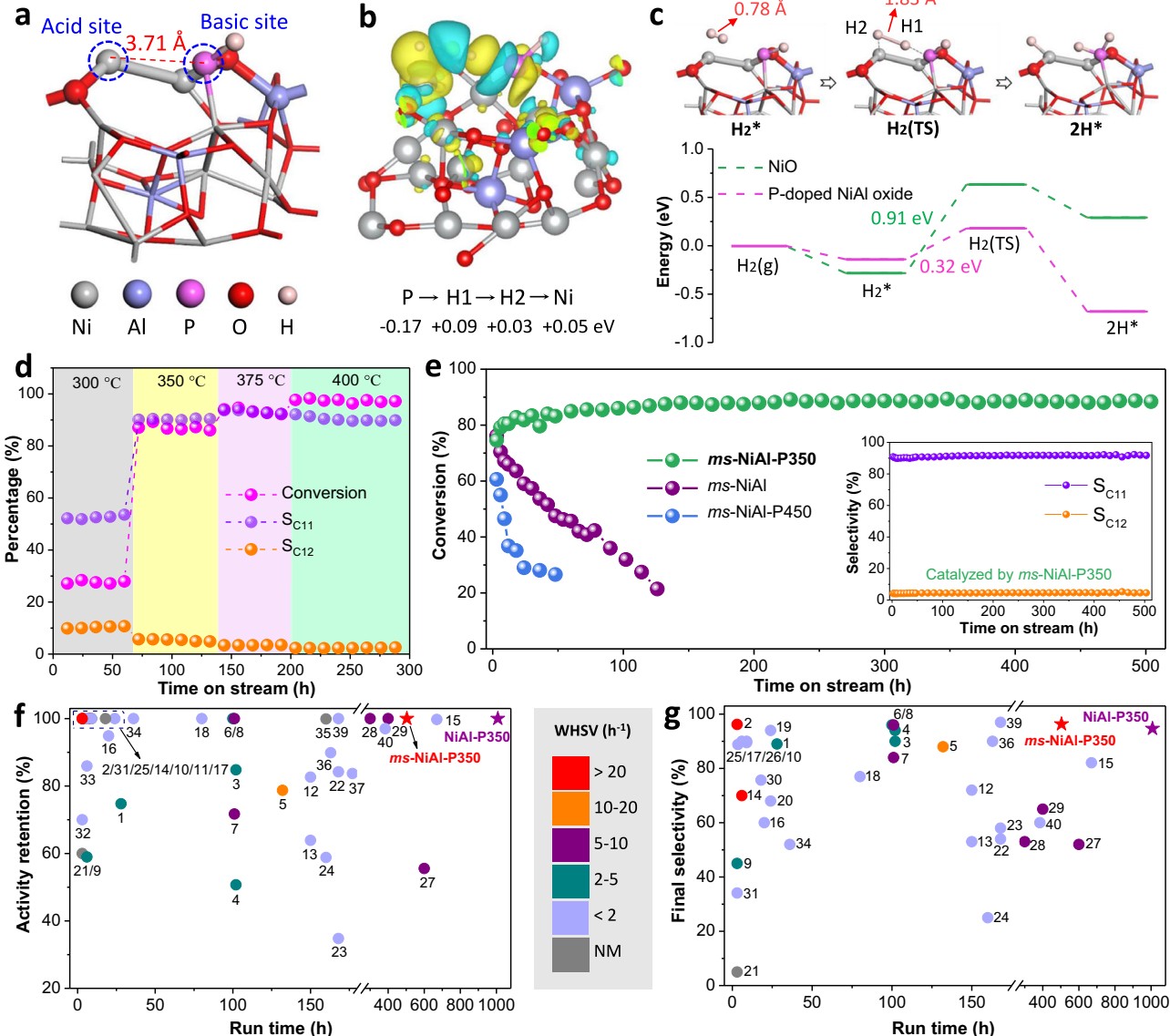

**Fig. 3 | DFT analysis and catalytic performance of *ms*-NiAl-P350. a** The proposed Ni(-O-Al)···P FLP in P-doped NiAl oxide model. **b** The differential charge density distribution of $H_2$ activation on the Ni(-O-Al)···P FLP (yellow isosurface means electron accumulation, blue isosurface means electron depletion). **c** The proposed $H_2$ activation process on the P-doped NiAl oxide model and the corresponding energy change profiles. **d** Temperature screening for hydrotreating of methyl laurate over *ms*-NiAl-P350. Reaction conditions: $P_{H2}$ = 3.0 MPa, WHSV = 28.3 $h^{-1}$. **e** Methyl laurate conversions as function of time over *ms*-NiAl-P350 and the

reference catalysts. The inset shows the selectivity of $C_{11}H_{24}$ and $C_{12}H_{26}$ over *ms*-NiAl-P350. Reaction conditions: T = 350 °C, $P_{H2}$ = 3.0 MPa, WHSV = 28.3 $h^{-1}$. **f, g** Comparison of *ms*-NiAl-P350 with other catalysts reported in literature: activity retention-time plots (**f**) and final selectivity-time plots (**g**). Numbers in (**f** and **g**) correspond to the row numbers in Supplementary Table 6, and the color of dots represents the WHSV used in the corresponding experimental system. NM means "not mentioned".

classical NiO Lewis pair (0.91 eV). It can therefore be anticipated that *ms*-NiAl-P350 with a well-defined FLP structure has the potential to catalyze the hydrogenation reactions.

## Catalytic performance in hydrogenation of methyl laurate

The distinctive structure of *ms*-NiAl-P350 and theoretical insights motivated us to investigate its catalytic properties in hydrotreating reaction using methyl laurate as a model compound in a fixed-bed continuous-flow reactor, wherein the target products were $C_{11}H_{34}$ and $C_{12}H_{26}$ (Supplementary Fig. 18)[20,33,34]. We screened the reaction temperature for methyl laurate hydrotreating over the *ms*-NiAl-P350 (reaction conditions: $P_{H_2}$ = 3.0 MPa, WHSV = 28.3 $h^{-1}$). After reaction at an initial temperature of 300 °C for 60 h, the temperature was raised gradually to 350 °C, 375 °C and 400 °C, and kept at each temperature for a certain duration (Fig. 3d). The conversion dramatically increased from 28% (300 °C) to 87% (350 °C), 94% (375 °C) and 98% (400 °C), respectively. The selectivity towards $C_{11}H_{24}/C_{12}H_{26}$ ($S_{C11+C12}$) increased from 63% (300 °C, the byproduct was mainly lauric acid) to 93–97% (350–400 °C, the oxygen was nearly fully removed). At each temperature, we observed no obvious deactivation during the reaction time period. In addition, we found that increasing $H_2$ pressure (ranging from 0.5 to 4 MPa) only slightly increased the methyl laurate conversion (Supplementary Fig. 19).

Considering the phase transformation of *ms*-NiAl-P350 (P-doped NiAl-oxide phase) to *ms*-NiAl-P450 ($Ni_2P$ phase), we set the hydrotreating temperature at 350 °C to investigate the long-term stability of *ms*-NiAl-P350 at WHSV of 28.3 $h^{-1}$ under $H_2$ pressure of 3.0 MPa. The *ms*-NiAl-P350 exhibited stable conversion of methyl laurate (85–88%) and selectivity towards $C_{11}H_{24}$ (~91.5%)/$C_{12}H_{26}$ (~4.5%) during 500 h of operation (stopped because of $H_2$ exhaustion) (Fig. 3e), far superior to the reference catalysts of *ms*-NiAl, *ms*-NiAl-P450 and *ms*-NiAl-$H_2$800-P350 (Fig. 3e and Supplementary Fig. 20), indicating its excellent durability. The NiAl-P350 synthesized without *ms*-template also showed good stability during 500 h of operation, but with a lower conversion (50–70%, Supplementary Fig. 21), probably due to the absence of hierarchical porous structure. We also compared the selectivity and stability of *ms*-NiAl-P350 with various catalysts reported in previous literature (Fig. 3f, g and Supplementary Table 6), and found that the *ms*-NiAl-P350 under relatively higher WHSV outperformed these catalysts (Supplementary Fig. 22). By decreasing the WHSV to <11.3 $h^{-1}$ (it is often set to 0.5 - 3.0 $h^{-1}$ in commercial system), a complete conversion with $S_{C11+C12}$ higher than 99.3% was achieved over the *ms*-NiAl-P350 (Supplementary Fig. 23).

## Mechanism analysis

In order to understand the stability mechanism of *ms*-NiAl-P350, we studied the spent catalysts in detail. The catalyst deactivation in hydrotreating processes is often caused by coke formation, metal agglomeration, phase transformation, component loss, etc[2,14,17,35]. In this work, the coke formation and P loss of *ms*-NiAl-P350 were negligible (Supplementary Fig. 24). After 24 h of operation, the *ms*-NiAl-P350-Sp.24 h well maintained its morphology (Fig. 4a-i), no obvious change in crystalline structure was observed (Fig. 4b). By comparison, metal agglomeration and phase transformation were clearly observed in the spent *ms*-NiAl-P450-Sp.24 h (Fig. 4a-ii, c and Supplementary Fig. 25). For the spent *ms*-NiAl-Sp.24 h, part of Ni species was transformed into Ni nanoparticles under high $H_2$ pressure (Fig. 4a-iii, d). Therefore, the *ms*-NiAl-P350 showed better stability under hydrotreating condition than *ms*-NiAl-P450 and *ms*-NiAl.

Prolonging the operation time to 500 h for *ms*-NiAl-P350, large Ni nanoparticles were formed (Fig. 4a-iv, b). Surprisingly, however, its catalytic performance was unaffected. The XPS spectra showed that, except for a slightly more pronounced peak being assigned to the metallic Ni (Fig. 4e), the chemical states of Ni, Al, and P species in *ms*-NiAl-P350-Sp.500 h remained unchanged compared with that of *ms*-NiAl-P350-Sp.24 h (Fig. 4e–g). Accordingly, the large metal particles were mainly composed of Ni species (Fig. 4h), rather than $Ni_xP_y$ nanoparticles, being in agreement with the XRD data (Fig. 4b). The remaining Ni element in *ms*-NiAl-P350-Sp.500 h was still mixed well with P, Al, and O species (Fig. 4h). Notably, we found no obvious P loss during 500 h of operation (Supplementary Fig. 24b). Therefore, we infer that the P-doped NiAl-oxide with FLP sites was quite stable under hydrotreating conditions in a long-term operation. The Ni species inside the NiAl-oxide skeleton without P-doping, which have less contribution to the catalytic performance, were prone to be reduced to form large Ni nanoparticles.

As mentioned above, fabricating the *ms*-NiAl ($Al^{3+}$-doped NiO-type NiAl-mixed oxide) by calcination at 800 °C under air was critical for the formation of P-doped NiAl-oxide. Lowering the calcination temperature (ranging from 650 to 350 °C) was not conducive to the formation of Ni–O–Al interaction, thus resulting in the formation of Ni–P bonds during phosphorization and poor catalytic performances (Supplementary Figs. 26–29). In addition, to confirm the necessity of FLPs in this catalytic system, we carried out a series of site-poisoning experiments. As known, pyrrole (strong Lewis acid) and pyridine (strong Lewis base) can be absorbed on the Lewis basic and acidic sites, respectively[36]. Therefore, the FLPs can be blocked by the pyridine and pyrrole molecules. It was found that after adding 20 wt% of pyrrole/pyridine into the catalytic system, the yield of target products decreased from 16.4% to 1.2% (pyrrole) and 2.1% (pyridine) at 300 °C, and from 82.0% to 33.6% (pyrrole) and 36.6% (pyridine) at 350 °C (Supplementary Fig. 30). These observations emphasized the necessity of Ni(-O-Al)···P FLPs construction for activating $H_2$ in order to achieve a high activity in the present catalytic system.

## Extension of catalytic system

To demonstrate the industrial viability of such a hydrotreating system, we tested the hydrotreating performance of the *ms*-NiAl-P350 using soybean oil as feedstock in the fixed-bed continuous-flow reactor (reaction conditions: T = 350 °C, $P_{H_2}$ = 3.0 MPa, WHSV = 6.0 $h^{-1}$). The *ms*-NiAl-P350 showed a stable activity with a nearly complete conversion of soybean oil and a selectivity towards C15-C18 hydrocarbons higher than 90% during 500 h of operation (Fig. 5a), implying its excellent durability, well surpassing the reported $Ni_2P/SiO_2$ catalyst (Fig. 5a and Supplementary Fig. 31, synthesized by the method of number 15 in Fig. 3f, g, see the Supplementary Materials for the experimental details). Moreover, we also tested various feedstocks in a continuous operation, including palm oil, duck fat, waste cooking oil, and soybean oil (Supplementary Table 7), all of which were completely converted with >90% selectivities towards C15-C18 hydrocarbons (Fig. 5b, c), indicating the outstanding adaptability of the *ms*-NiAl-P350 to various feedstocks.

Furthermore, we prepared the bulk NiAl-P350 catalyst (without *ms*-template) on a kilogram scale and shaped it into extrudates (Supplementary Fig. 32), over which a stable conversion of methyl laurate (~80%) with high selectivity towards $C_{11}H_{24}$ (~90%)/$C_{12}H_{26}$ (~4%) was achieved during 1000 h of operation at WHSV of 9.4 $h^{-1}$. However, the catalytic reaction rate of bulk NiAl-P350 is significantly lower than that of *ms*-NiAl-P350 (Supplementary Fig. 33). This could be attributed to the reduced porosity and surface area of bulk NiAl-P350, which limit the internal mass transfer and accessibility of active sites (Supplementary Fig. 34). It should be noted that the two catalysts possess similar chemical characteristics as determined by the HRTEM, XRD and XPS analysis (Supplementary Figs. 35 and 36). This comparison demonstrates that replicating the biological structure of *ms* to generate hierarchical pores is a highly effective approach to enhance the catalytic performance of such FLP-type catalysts. However, the bulk NiAl-P350 would be a suitable alternative for industrial applications, taking into account the expense and magnitude of catalyst production.

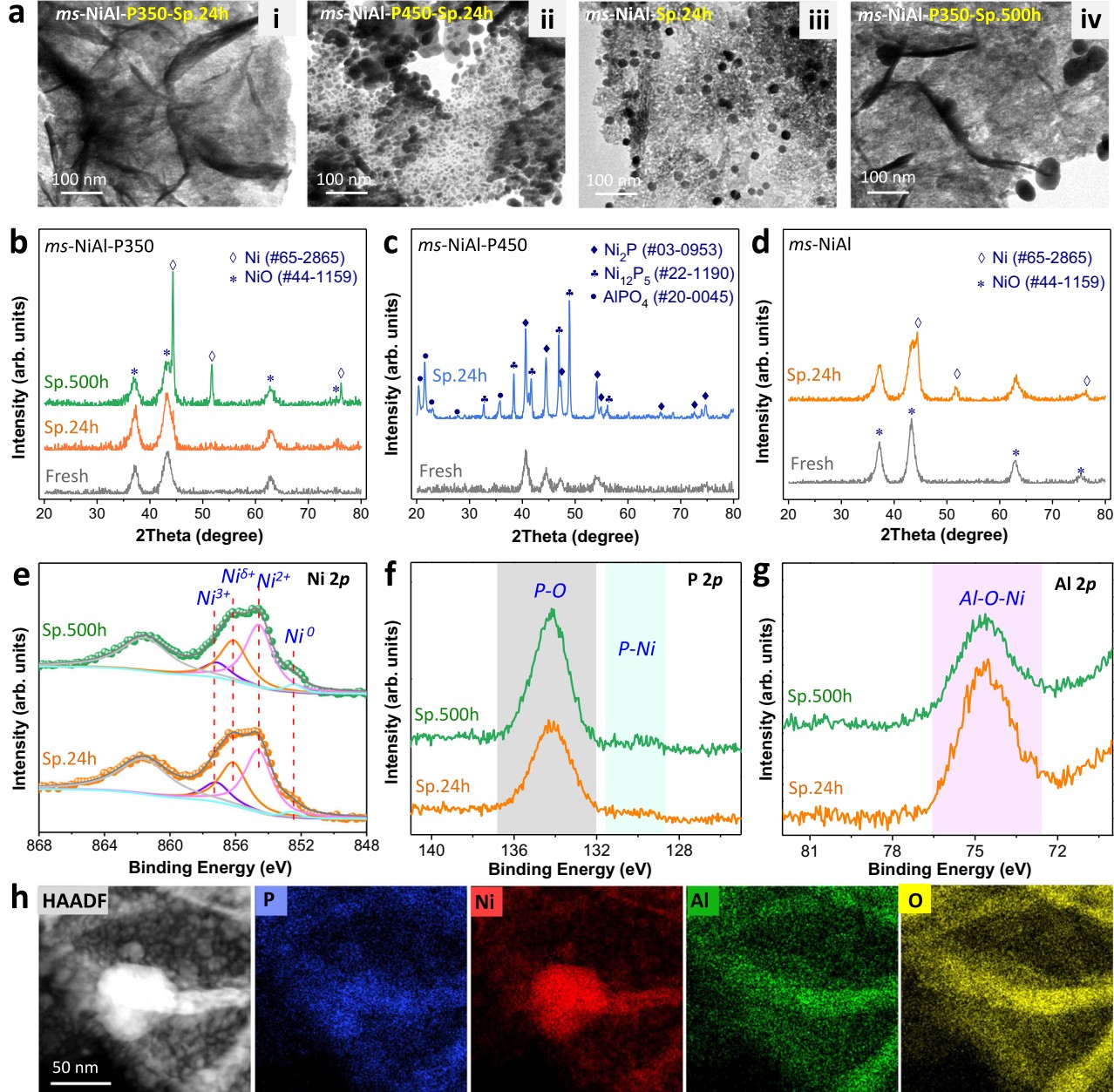

**Fig. 4 | Investigation of the stability mechanism of *ms*-NiAl-P350. a** TEM images of the spent catalysts. **b–d** Comparison of XRD patterns of the fresh and spent catalysts. **e–g** XPS spectra of the spent *ms*-NiAl-P350 after 24 h and 500 h of operations. **h** HAADF-STEM and EDS elemental mapping images of the spent *ms*-NiAl-P350 after 500 h of operation. The large metal particles mainly contain Ni species. The P, Al, O, and Ni (except the Ni particles) elements in the spent *ms*-NiAl-P350 are still mixed well with one another.

## Discussion

In conclusion, we have developed the P-doped NiAl-oxide as a robust catalyst for efficient production of green diesel from renewable oils. The catalyst demonstrates outstanding durability and completely converts various feedstocks, e.g., methyl laurate, soybean oil, palm oil, duck fat, and waste cooking oil, with >90% selectivity towards diesel-range hydrocarbons. Notably, there is no need to replenish the sulfur source or other species during hydrotreating process, making the green diesel production much greener and cheaper. We demonstrated that the benign interaction between the mixed NiAl-oxide and P (i.e., Ni(-O-Al)···P FLP) contributed to the outstanding catalytic performance, and the synthesis of Ni(-O-Al)···P FLP-type catalyst can be readily scaled up in kilogram level. Moreover, the structural and compositional flexibility of LDHs (precursor of the P-doped NiAl-oxide

catalyst) offers immense opportunities to prepare other FLP-type catalysts for a variety of catalytic reactions beyond green diesel production.

## Methods
### Catalyst preparation
**Synthesis of *ms*-NiAl-LDHs.** Typically, 4.5 mmol of Ni(NO$_3$)$_2$·6H$_2$O, 1.5 mmol of Al(NO$_3$)$_3$·9H$_2$O, and 13.5 mmol of urea were dissolved in 90 mL of water. Then, the solution together with 1.0 g of the pretreated *ms* was transferred into a round bottom flask, followed by vacuum treating for three times to ensure full filling of the *ms* cells by the solution containing precursors. Next, the mixture was transferred into a teflon-lined stainless-steel autoclave, sealed and heated up to 100 °C, and kept at this temperature for 24 h. The product was collected by

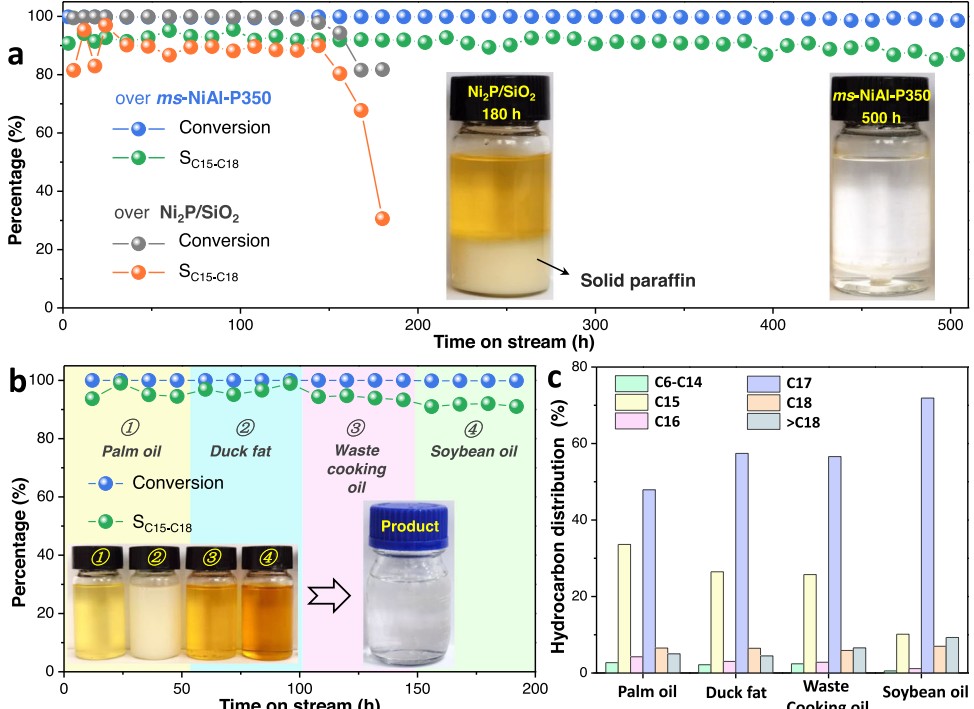

**Fig. 5 | Hydrotreating of various oils. a** Soybean oil conversion and selectivity towards C15-C18 hydrocarbons (inset, photographs of the products) over *ms*-NiAl-P350 and Ni₂P/SiO₂ (synthesized according to the method of number 15 in Fig. 3f, g). **b** Catalytic performances of *ms*-NiAl-P350 using various oils as feedstocks during continuous operation (inset, photographs of the feedstocks and the products) and (**c**) the corresponding distributions of hydrocarbon products. Reaction conditions: T = 350 °C, $P_{H2}$ = 3.0 MPa, WHSV = 6.0 h⁻¹.

filtration, washed several times with water and finally freeze-dried (*ms*-NiAl-LDHs).

**Synthesis of *ms*-NiAl-P350.** Firstly, the *ms*-NiAl-LDHs was placed in a muffle furnace, calcined at 800 °C for 1 h, and then naturally cooled down to room temperature. The biological structured NiAl-mixed oxide was obtained (*ms*-NiAl). Then, the NaH₂PO₂ powder and the as-obtained *ms*-NiAl were placed into a quartz tube (separated by quartz cotton), treated at 150 °C for 30 min in an Ar flow (30 mL/min). The quartz tube was pumped to form a vacuum, sealed, heated to 350 °C, and held at this temperature for 2 h. After cooling down to room temperature, the sample was passivated in a flow of 0.5% O₂ in Ar (100 mL/min) for 2 h (*ms*-NiAl-P350).

**Synthesis of Ni₂P/SiO₂.** The Ni₂P/SiO₂ was prepared according to the procedure described in literature[25]. Briefly, the SiO₂ (S817570, Macklin, BET = 300 m²/g) was impregnated with an aqueous solution of Ni(NO₃)₂·6H₂O and NH₄H₂PO₄ (mass ratio of SiO₂/Ni(NO₃)₂·6H₂O/NH₄H₂PO₄ is 1/1.6/0.45) stabilized with HNO₃ (keeping the pH in the range of 4–4.5 to prevent precipitate formation). Then, the sample was dried at 120 °C for 4 h and calcined at 500 °C for 6 h. The as-obtained powder was pelletized, crushed, sieved to a fraction of 25–60 mesh, and then reduced at 580 °C for 30 min in H₂ flow to form Ni₂P phase (Ni₂P/SiO₂).

### Catalytic tests
The hydrotreating process was carried out in a continuous-flow stainless-steel fixed-bed reactor (inner diameter of 8 mm). In the blank reactor, the conversion of methyl laurate was negligible (<1%, even at 400 °C). Typically, 100 mg of *ms*-NiAl-P350 was cut into pieces, loaded into the reactor, and held by quartz cotton from the upper and lower sides. The catalyst was in situ treated at 350 °C for 2 h in a H₂ flow (>99.9%, 118 mL/min). Then, the H₂ pressure was adjusted to 3.0 MPa,

and the methyl laurate was continuously fed into the reactor by using a liquid micro pump. The molar ratio of H₂/methyl laurate was set as 25, and the weight hourly space velocity (WHSV) of methyl laurate was 28.3 h⁻¹. The liquid products were identified by GC-MS (Agilent 7890B-5977B) and quantitatively analyzed by GC equipped with FID detector (Agilent 7890B). Tetrahydronaphthalene was used as an internal standard. The gaseous products were quantitatively analyzed by on-line GC equipped with a TCD and a TDX-101 packed column. N₂ was used as an internal standard. The mass balance values were always higher than 95% for all the test points. The methyl laurate and soybean oil were used directly as a feedstock without dilution. The palm oil, duck fat, and waste cooking oil were diluted with an equal mass of hexane before using, as they exhibit poor fluidity at room temperature. The conversion and selectivity towards hydrocarbons were calculated using the following equations:

$$\text{Conversion}\,(\%) = \frac{\text{moles of feedstock in feed} - \text{moles of feedstock in product}}{\text{moles of feedstock in feed}} \times 100\%$$

$$\text{Selectivity}\,(\%) = \frac{\text{total moles of hydrocarbons in product}}{\text{total moles of fatty acids in feed} \times \text{conversion}} \times 100\%$$

### Characterization methods
A field-emission scanning electron microscope (SEM, Hitachi S4800) was employed to investigate the surface morphology. Transmission electron microscopy (TEM) images were acquired with Hitachi-7650 microscopes. High-resolution transmission electron microscope (HRTEM, Talos F200S, FEI) was adopted to analyze the internal structure and element mapping of catalysts. X-ray diffraction (XRD) patterns were acquired using a Bruker D8 Advance X-ray diffractometer with Cu Kα radiation (λ = 1.54 Å). X-ray photoelectron spectroscopy (XPS) was recorded with an X-ray photoelectron spectrometer (ESCALAB250, Thermo-VG Scientific, UK). Elemental

contents were determined by an inductively coupled plasma-optical emission spectrophotometer (ICP-OES, 730 Series, Agilent, USA). The pore structure of samples was determined by $N_2$ sorption isotherm experiments at 77 K with a Micromeritics Gemini apparatus (Tristar II 3020 M, Micromeritics Instrument Co., USA). The specific surface area was calculated with the Brunauer-Emmett-Teller (BET) method, and pore volume was determined by Barrett-Joiner-Halenda (BJH) method. The thermogravimetric analysis was conducted with a thermogravimetric analyzer (TGA, SDT Q600, TA Instruments, USA) equipped with a thermal detector (DTG-60H). For TGA analysis, a certain amount of sample was placed in an alumina pan and degassed with Ar or $H_2$/Ar (10%/90%), then heated from an ambient temperature to 800 °C with a heating rate of 10 °C/min. The temperature programmed reduction (TPR) analysis was conducted with an automatic chemical adsorption analyzer (Micromeritics AutoChem II 2920). For TPR experiments, 20 mg catalyst was loaded into a quartz tube and heated in 50 mL/min of 10 vol.% $H_2$/Ar at a rate of 10 °C/min.

The X-ray absorption fine structure (XAFS) measurements were performed at the beamline 1W1B of the Beijing Synchrotron Radiation Facility (BSRF), Institute of High Energy Physics (IHEP), Chinese Academy of Sciences (CAS). Extended X-ray absorption fine structure spectra (EXAFS) were recorded at ambient temperature in transmission mode. The typical energy of the storage ring was 2.5 GeV with a maximum current of 250 mA; the Si (111) double crystal monochromator was used. Fourier transform of the EXAFS spectra was carried out in a K-range from 3.0 to 12.8 Å$^{-1}$. The IFFEFIT 1.2.11 date analysis package (Athena, Artemis, Atoms, and FEFF6) was used for the date analysis and fitting.

Synchrotron X-ray tomography was carried out at the P05 beamline at DESY, Hamburg, Germany. For the measurements conducted here, the synchrotron beam energy was monochromatized to 20 KeV using a double multilayer monochromator (DMM). A $CdWO_4$ single crystal scintillator of 100 μm thickness was used to convert the X-ray to visible light. A fast KIT CMOS camera (5120 × 3840 pixels) that was kept out of the direct beam by using a mirror was used. 2400 projections within a 180° battery rotation were recorded with the exposure time of 0.16 s. The field of view is 3.28 × 2.46 mm$^2$, with a pixel size of ~0.6 μm. Note that for the cells measured at the P05 beamline, a binning process of 2 by 2 was used when normalizing the dataset to get a high signal-to-noise ratio. As a result, the resultant spatial resolution is 1.2 μm. The tomography data from P05 beamline were reconstructed using beamline Matlab ASTRA Toolbox. The analysis of the tomography data was conducted by Avizo.

### Computational methods

The Vienna Ab Initio Package (VASP) was employed to perform all the spin-polarized density functional theory (DFT) calculations within the generalized gradient approximation (GGA) in the PBE formulation. The projected augmented wave (PAW) potentials was chosen to describe the ionic cores and take valence electrons into account using a plane wave basis set with a kinetic energy cutoff of 400 eV. Partial occupancies of the Kohn–Sham orbitals were allowed using the Gaussian smearing method and a width of 0.05 eV. The electronic energy was considered self-consistent when the energy change was smaller than $10^{-5}$ eV. A geometry optimization was considered convergent when the residual forces were smaller than 0.02 eV/Å. Grimme's DFT-D3 methodology was used to describe the dispersion interactions.

The equilibrium lattice constant of NaCl-type NiO was optimized, when using a $11 \times 11 \times 11$ Monkhorst-Pack k-point grid for Brillouin zone sampling, to be $a = 4.418$ Å. We then use it to construct a NiO(001) monolayer model (model 1) with $p(2 \times 2)$ periodicity in the x and y directions and 2 stoichiometric layers in the Z direction together with a vacuum layer in the Z direction in the depth of 15 Å in order to mitigate the interactions between the slab and its periodic images. Model 1 comprises of 32 Ni and 32 O atoms. Model 2 was built by replacing 8 Ni atoms in model 1 with Al atoms. Model 3 was built by adding a -PH group onto model 2. During structural optimizations, a $3 \times 3 \times 1$ k-point grid in the Brillouin zone was used for k-point sampling, and the bottom stoichiometric layer was fixed while the top one was allowed to relax.

The adsorption energy ($E_{ads}$) of adsorbate A was defined as

$$E_{ads} = E_{A/Surf} - E_{Surf} - E_A(g)$$

where $E_{A/Surf}$, $E_{Surf}$, and $E_A(g)$ are the energy of adsorbate A adsorbed on the surface, the energy of clean surface, and the energy of isolated A molecule in a cubic periodic box with a side length of 20 Å and a $1 \times 1 \times 1$ Monkhorst-Pack k-point grid for Brillouin zone sampling, respectively.

The $H_2$ dissociation energy was defined as

$$E = 2E_{H^*} - E_{H2^*}$$

where $E_{H^*}$ and $E_{H2^*}$ are the energy of an adsorbed H atom, the energy of an adsorbed $H_2$ molecule, respectively.

The transition state of an elementary reaction step was located by the nudged elastic band (NEB) method. In the NEB method, the path between the reactant(s) and product(s) was discretized into a series of five structural images. The intermediate images were relaxed until the perpendicular forces were smaller than 0.02 eV/Å.

## Data availability

All relevant data that support the findings of this study are presented in the manuscript and supplementary materials. Source data are available from the corresponding author upon reasonable request. Source data are provided with this paper.

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

## Acknowledgements

The work was supported by the National Natural Science Foundation of China (52100169, 22008249) and Natural Science Foundation of Shandong Province China (ZR2020QB196). We acknowledge Helmholtz-Zentrum Hereon (Geesthacht, Germany) for provision of the PETRA III P05 beamline and conducting the synchrotron X-ray tomography measurements as well as DESY (Hamburg, Germany) for operating the PETRA III synchrotron facility. We thank Fu Sun, Ingo Manke, Markus Osenberg, Fabian Wilde, and André Hilger for conducting the synchrotron X-ray tomography measurements and reconstructing and analyzing the tomography datasets.

## Author contributions

D.C.L. carried out the preparation and characterization of catalysts, tested the catalytic performances, analyzed the data, prepared the Figures, and drafted the manuscript. Z.P. provided the method of catalytic experiments. Z.T., Q.Z., X.D. and H.J. participated in the data analysis. G.H.W. initiated and supervised the project and paper writing. All the authors approved the final version of the manuscript.

## Competing interests

Three patent applications related to this work were filed by Qingdao Institute of Bioenergy and Bioprocess Technology, Chinese Academy of Sciences. G.H.W., D.C.L. and Z.P. are inventors of a China patent (patent number: ZL202110886367.8) covering the core technology of catalyst preparation and its application in hydrodeoxygenation reactions. G.H.W. and D.C.L. are inventors of a China patent (patent number: ZL202110955550.9) covering the preparation of catalyst with biological architecture. The above two patents have been granted by China National Intellectual Property Administration. G.H.W., D.C.L., and Z.P. are inventors of an international patent (application number: PCT/CN2021/130970) covering the core technology of catalyst preparation and its application in hydrodeoxygenation reactions, and the patent is currently in the national stage (USA and European region). All other authors declare no competing interests.
