## [Peer Review File · Nature Communications]

Frustrated Lewis pair catalyst realizes efficient green diesel productionREVIEWER COMMENTS

Reviewer #1 (Remarks to the Author):

The manuscript reports on a novel catalytic system, based on P-doped NiAl oxides prepared using maize straw as template, which shows outstanding catalytic properties for the production of green diesel by hydrotreatment of a variety of biomass-derived oil substrates. One of the most relevant results is the remarkable activity of this system along the time on stream, with almost no variations being observed during tests up to 500 h. Moreover, the tests are performed under sulfur-free conditions which represents a great advantage compared with conventional/commercial catalysts. The authors make also a detailed comparison with previous literature showing the relevance of the results here reported.

This catalyst system, and several reference samples, are fully characterized by a large variety of conventional and advanced techniques, concluding that the origin of this singular catalytic performance is the generation of so called frustrated Lewis pairs coming from P-O-Ni sites. Parts of the manuscript are somewhat confusing, mainly because a huge amount of experimental data are discussed in a short text in a superficial way. In this way, the supporting information includes a total of 43 figures and 7 tables. Some of the figures contain duplicate information. My suggestion is to include the methods section as supporting information, using the space so available for improving and extending the discussion of the results, as well as to remove unnecessary or duplicate information. Specific comments:

- Textural properties of the catalyst samples are not provided. Is there any reason for that?
- As the own authors indicate, the catalytic system prepared without using the maize straw template exhibits also remarkable catalytic properties: somewhat lower conversion but very high stability along the time on stream. This could benefit significantly the economy of the process, reducing the catalyst cost and making its preparation simpler. This issue should be assessed and commented.

Reviewer #2 (Remarks to the Author):

In this submission from Li and co-workers, a series of Ni-Al-P oxide materials was prepared, and the conditions leading to a P-alloyed oxide, rather than Ni₂P, was identified. A biological structure directing agent was utilized to prepare high surface area and porous Ni-Al LDH, which is an interesting approach. The materials were tested as catalysts for hydrotreating of various model compounds and real feedstocks. While this reviewer appreciates the combination of fundamental and applied efforts, the impact delivered from this approach do not meet the standards required for Nature Communications. The following are points where the manuscript does not bring the needed impact, and are thoughts for the authors to consider when resubmitting the manuscript elsewhere:

(1) The authors motivate this report, and repeat throughout, that the results here have the potential to disrupt the multi-billion dollar sulfided catalyst industry. I disagree. It would take an extremely special material to disrupt this industry, including one that is especially easy to prepare. I appreciate the materials chemistry that went into the "ms" porous structure reported here, but that is not a scalable material, requiring quite high temperatures and phosphorus sources that are on regulated chemicals lists. Further, alternatives to sulfided materials that are based on P-containing catalysts must overcome a similar barrier of "P maintenance", analogous to S maintenance. It is clear that the catalysts reported here change phase after hundreds of hours of reaction, suggesting that some type of P co-feed may be necessary, and this is a significant challenge to commercialization.

(2) The combination of applied and fundamental approaches here reads as the combination of 2 separate papers, but these do not bring a high impact result. The results presented from the bulk P-Ni-Al oxide (Figure 5), which is the more scalable version of the catalyst, demonstrate that all the materials chemistry associated with the porous material is not needed and the whole focus of the

paper on that material is unnecessary - it should be focused on this more scalable material. From the applied catalysis perspective, the catalytic data was not presented with the appropriate level of rigor to show that the "ms" version of the catalyst provides any advantage other than surface area. That is, there are no site density measurements that would enable a turnover frequency analysis, which would show that the 2 catalyst materials are operating the same or differently.

(3) Thus, this combination of results would be better served as 2 separate papers: one focused in an applied catalysis journal where the preparation, characterization, and catalytic data from the non-structured, bulk material is presented. This would enable stronger arguments about the potential to replace sulfided catalysts. Then, a second paper could focus on the interesting materials chemistry using a structure directing agent that enables the production of high surface area analogs to these bulk materials. Some metric of performance must be found that differentiates these materials beyond an increase surface area, and perhaps further structural analysis would find this.

Reviewer #3 (Remarks to the Author):

In this submission, the authors reported a frustrated Lewis pair catalyst to realize efficient green diesel production, and investigated its charge transfer mechanism in H₂ activation. However in my opinion, the results and analysis is not sufficient to support their principle and conclusion.

Some of my questions are as below:

1. For the frustrated Lewis pair of Ni(-O-Al)•••P, the more positive Ni site caused by electron transfer from Ni-O to the nearby O-Al is an extremely important factor. The authors demonstrated this electron transfer by the higher white line intensity of Ni K-edge, and the existence of Ni^{δ+} ($2 < \delta < 3$) in Ni 2p_{3/2} XPS spectra. However, the more important and direct evidence is lacked, which is the comparison in Ni and Al XPS spectra between pure NiO or Al₂O₃ and Al-doped NiO.
2. There're some serious problems in the (P-doped) NiAl-oxide models used in DFT calculation.
 - (a) In model 2, why is the crystal structure of the top Ni-Al-O layer is vastly different from that of the bottom layer? The top layer seems to be close to amorphous nature.
 - (b) According to P 2p XPS spectra, the peak at 133.8 eV in ms-NiAl-P350 is arises from P-O species, such as PO₃, PO₄, P₂O₅, etc. In the model 3, the PH group is bonding to one O atom. I don't think this P-relevant construction is accurately representative for its actual structure reflected by XPS. BTW, the a PH₂ group was used in model 3 of Supplementary Fig. 17 by mistake.
 - (c) According to the elemental contents of ms-NiAl-P350 revealed by ICP-OES (0.44 P/Ni atomic ratio) or XPS (0.91 P/Ni atomic ratio), the P/Ni atomic ratio in model 3 differs too much. So the sufficient P doping in bulk should be considered in DFT model, not just the PH group on the surface.
3. The analysis in electron transfer between adsorbed H₂ and FLP site by Bader charges is confused. Although the positive charge of Ni decreased (0.697 to 0.645), the electron of H₂ atom bonded to Ni also increased (-0.012 to -0.039). So the 0.05 electron transfer from H₂ to Ni is not convictive enough. The increased electron of Ni may be originated from its other coordination atoms (e.g. O), rather than H₂. That means the role of Ni acid site in FLP sites is not well established. The authors should ascertain it by more careful investigation.
4. To demonstrate the advantage of FLP site, the catalytic abilities of contrast sample with single acid site (ms-NiAl) or basic site (P-doped NiO) also should be investigated. Likewise, the energy change profiles of H₂ activation process on catalysts with single acid/basic site are needed, to confirm the promotion in H-H scission by FLP site in comparison with single site catalysts.
5. As shown in Fig 3d, the conversion of methyl laurate kept at 85-88%, with selectivity of C₁₁+C₁₂ at 91.5%+4.5% in the long-term running of ms-NiAl-P350 (350 °C, WHSV=28.3 h⁻¹) However in Fig 4d, the conversion (red star) is quite closed to 100%.

Response to Reviewers' Comments

Response to Reviewer 1

Referee Letter:

The manuscript reports on a novel catalytic system, based on P-doped NiAl oxides prepared using maize straw as template, which shows outstanding catalytic properties for the production of green diesel by hydrotreatment of a variety of biomass-derived oil substrates. One of the most relevant results is the remarkable activity of this system along the time on stream, with almost no variations being observed during tests up to 500 h. Moreover, the tests are performed under sulfur-free conditions which represents a great advantage compared with conventional/commercial catalysts. The authors make also a detailed comparison with previous literature showing the relevance of the results here reported.

This catalyst system, and several reference samples, are fully characterized by a large variety of conventional and advanced techniques, concluding that the origin of this singular catalytic performance is the generation of so called frustrated Lewis pairs coming from P-O-Ni sites.

Parts of the manuscript are somewhat confusing, mainly because a huge amount of experimental data are discussed in a short text in a superficial way. In this way, the supporting information includes a total of 43 figures and 7 tables. Some of the figures contain duplicate information. My suggestion is to include the methods section as supporting information, using the space so available for improving and extending the discussion of the results, as well as to remove unnecessary or duplicate information.

Response: We appreciate the reviewer's encouraging assessment and helpful suggestions. Following the reviewer's suggestion, we have integrated the interrelated figures and removed any possible duplicate information from the section of supplementary materials to improve the concision of data. Meanwhile, we have extended the discussion of experimental results in the revised manuscript and supplementary materials to improve the readability. As for the methods section, we keep it in the main article following the editor's suggestion. The main changes we have made in the revised version are as follows:

- The previous Supplementary Fig. 11 (elemental mapping images of *ms*-NiAl-P350), 32 (elemental mapping images of *ms*-NiAl-P450-Sp.24h), 39 (atomic ratios of P/Ni

for *ms*-NiAl(t)-P350) and Table 7 (elemental contents of products) have been deleted.

- The previous Supplementary Fig. 1 and 2 have been integrated as Supplementary Fig. 1 in the revised version.
- The previous Supplementary Fig. 3 and 4 have been integrated as Supplementary Fig. 2 in the revised version.
- The previous Supplementary Fig. 12 and 13 have been integrated as Supplementary Fig. 9 in the revised version.
- The previous Supplementary Fig. 14 and 15 have been integrated as Supplementary Fig. 10 in the revised version.
- The previous Supplementary Fig. 24 and 25 have been integrated as Supplementary Fig. 20 in the revised version.
- The previous Supplementary Fig. 29 and 30 have been integrated as Supplementary Fig. 24 in the revised version.
- The previous Supplementary Fig. 31, 33 and 34 have been integrated as Supplementary Fig. 25 in the revised version.
- The previous Supplementary Fig. 35, 36 and 38 have been integrated as Supplementary Fig. 26 in the revised version.
- The previous Supplementary Fig. 37 and 42 have been integrated as Supplementary Fig. 29 in the revised version.
- The synthesis details of *ms*-NiAl-H₂800-P350 have been added following Supplementary Fig. 11.
- A detailed discussion about the effect of calcination temperature (Supplementary Fig. 26-29) has been added in the revised version.

In the revised version, the supporting information includes 34 figures and 7 tables.

Specific comments:

1. Textural properties of the catalyst samples are not provided. Is there any reason for that?

Response: We thank the reviewer for this question. The textural properties of samples have been provided in Supplementary Table 3, and the discussion has been added to the manuscript on page 6.

“The Brunauer-Emmett-Teller (BET) surface area of ms-NiAl-P350 is $103.4 \text{ m}^2 \cdot \text{g}^{-1}$, which is larger than that of NiAl-P350 powder ($82.4 \text{ m}^2 \cdot \text{g}^{-1}$, synthesized without ms-template, Supplementary Fig. 7). Due to the high porosity, the ms-NiAl-P350 exhibited a much lower apparent density of $11.7 \text{ mg} \cdot \text{cm}^{-3}$ (Fig. 1a, Supplementary Fig. 8 and Supplementary Table 3) than the $307.1 \text{ mg} \cdot \text{cm}^{-3}$ of NiAl-P350 powder.”

Supplementary Table 3. Textural properties of samples.

Sample	BET surface Area ($\text{m}^2 \cdot \text{g}^{-1}$)	Pore Volume ($\text{cm}^3 \cdot \text{g}^{-1}$)	Average pore width (nm)	Apparent density ($\text{mg} \cdot \text{cm}^{-3}$)
NiAl powder	134.8	0.6	18.7	232.3
ms-NiAl	248.9	0.7	12.9	9.6
NiAl-P350 powder	82.4	0.5	23.8	307.1
ms-NiAl-P350	103.4	0.4	17.7	11.7
bulk NiAl-P350	78.3	0.3	14.0	2770.0

2. As the own authors indicate, the catalytic system prepared without using the maize straw template exhibits also remarkable catalytic properties: somewhat lower conversion but very high stability along the time on stream. This could benefit significantly the economy of the process, reducing the catalyst cost and making its preparation simpler. This issue should be assessed and commented.

Response: We would like to thank the reviewer for this constructive suggestion. Considering the catalyst cost and easier preparation, it is indeed more economical to produce the bulk NiAl-P350. In the revised version, we have added the data about the characterization of bulk NiAl-P350, its catalytic activity in comparison to ms-NiAl-P350, and the comments of the bulk NiAl-P350 catalyst.

“Furthermore, we prepared the bulk NiAl-P350 catalyst (without ms-template) on a kilogram scale and shaped it into extrudates (Supplementary Fig. 32), over which a stable conversion of methyl laurate (~80%) with high selectivity towards $\text{C}_{11}\text{H}_{24}$

(~90%)/C₁₂H₂₆ (~4%) was achieved during 1000 h of operation at WHSV of 9.4 h⁻¹. The reaction efficiency over the bulk NiAl-P350 catalyst (36.4 mmol·g⁻¹·h⁻¹) is lower than that over the ms-NiAl-P350 catalyst (117.9 mmol·g⁻¹·h⁻¹), which should be attributed to its limited accessibility of active sites, given that the two catalysts possess similar chemical characteristics as determined by the HRTEM, XRD and XPS analysis (Supplementary Figs. 33 and 34). This comparison demonstrates that replicating the biological structure of ms to generate hierarchical pores is a highly effective approach to enhance the catalytic performance of such FLP-type catalysts. However, the bulk NiAl-P350 would be a suitable alternative for industrial applications, taking into account the expense and magnitude of catalyst production.” (Page 16)

Supplementary Fig. 33. HAADF-STEM and the corresponding EDS elemental mapping images of bulk NiAl-P350.

Supplementary Fig. 34. (a) XRD patterns of bulk NiAl and bulk NiAl-P350. (b-d) XPS spectra of bulk NiAl-P350.

Response to Reviewer 2

Referee Letter:

In this submission from Li and co-workers, a series of Ni-Al-P oxide materials was prepared, and the conditions leading to a P-alloyed oxide, rather than Ni₂P, was identified. A biological structure directing agent was utilized to prepare high surface area and porous Ni-Al LDH, which is an interesting approach. The materials were tested as catalysts for hydrotreating of various model compounds and real feedstocks. While this reviewer appreciates the combination of fundamental and applied efforts, the impact delivered from this approach do not meet the standards required for Nature Communications. The following are points where the manuscript does not bring the needed impact, and are thoughts for the authors to consider when resubmitting the manuscript elsewhere:

Response: We thank the reviewer for the assessment of our manuscript. In this work, we designed a “*ms*” porous structured FLP catalyst, which enables efficient green diesel production without replenishing the sulfur source or other species during the hydrotreating process. We have demonstrated that the FLP structure plays a crucial role in achieving coke- and sintering-resistant activity during the hydrotreating process. To the best of our knowledge, such FLP configuration has never been reported before. So, we believe that our findings represent important advances in the fields of heterogeneous catalysis, materials science, and renewable energy, which should meet the standards required for *Nature Communications*. Below is a point-by-point response to the reviewer, which explains why we think it is suitable for publication in *Nature Communications*.

1. The authors motivate this report, and repeat throughout, that the results here have the potential to disrupt the multi-billion dollar sulfided catalyst industry. I disagree. It would take an extremely special material to disrupt this industry, including one that is especially easy to prepare. I appreciate the materials chemistry that went into the "ms" porous structure reported here, but that is not a scalable material, requiring quite high temperatures and phosphorus sources that are on regulated chemicals lists.

Response: We would like to express our gratitude to the reviewer for recognizing our efforts regarding the “*ms*” porous structure. In fact, constructing the “*ms*” porous structure can significantly improve the catalytic performance, resulting in a reaction efficiency ($117.9 \text{ mmol} \cdot \text{g}^{-1} \cdot \text{h}^{-1}$) that is three times higher than that of a catalyst without the “*ms*” structure ($36.4 \text{ mmol} \cdot \text{g}^{-1} \cdot \text{h}^{-1}$). From another perspective, the use of “*ms*” as a sacrificial biotemplate for shaping highly active catalysts may provide a good way for utilization of *ms*-waste (global annual yield of ~1.1 billion tons).

During the synthesis of a P-doped NiAl-oxide catalyst, two steps that require high temperature are involved, namely calcination at 800 °C and phosphorization at 350 °C. These processes are similar to those used in the industrial production of sulfided catalysts (calcination at 500-800 °C and sulfuration at 200-350 °C). The phosphorus source used in this study is sodium hypophosphite (NaH_2PO_2), which has been widely

used in the industrial field and is not listed as a regulated chemical in China. So, the production of this particular type of catalyst on a large scale would be easily accomplished under the present industrial conditions.

In this manuscript, we aimed to introduce the synthesis of a “*ms*” porous structured FLP catalyst and to describe its catalytic nature in green diesel production. These contents are important in the fields of materials science and heterogeneous catalysis, and are expected to be suitable for a broad readership of *Nature Communications*.

We completely agree with the reviewer that it would take an extremely special material to disrupt this industry, including one that is especially easy to prepare. Taking this issue into account, the catalyst without “*ms*” template would be a suitable alternative for industrial applications. Indeed, this is the reason for including non-*ms* catalyst data at the end of this manuscript, which serves as further evidence to prove the industrial viability of such a FLP catalyst and also draws the attention of readers from the industry. We think that this could make the manuscript more suitable for publication in the journal of *Nature Communications*, which has a broadly interested readership.

2. Further, alternatives to sulfided materials that are based on P-containing catalysts must overcome a similar barrier of "P maintenance", analogous to S maintenance. It is clear that the catalysts reported here change phase after hundreds of hours of reaction, suggesting that some type of P co-feed may be necessary, and this is a significant challenge to commercialization.

Response: We thank the reviewer for this important comment. In fact, it is unnecessary to add P sources to our reaction system, which is a noteworthy advantage of our catalyst in comparison to sulfided catalysts. According to the ICP analysis (Supplementary Fig. 24b and Table 1), although the atomic ratio of P/Ni in *ms*-NiAl-P350 decreased during the initial 24 h (possibly due to the loss of physically adsorbed P species), it is worth noting that the P/Ni ratio remained nearly unchanged throughout the remaining reaction duration. This indicates that there is no obvious P loss during 500 h of operation.

Supplementary Fig. 24. (b) Atomic ratios of P/Ni for fresh and spent *ms*-NiAl-P350 determined by ICP-OES.

As the reviewer mentioned, the catalyst changes phase after 500 h reaction, and large Ni nanoparticles were observed in the sample of *ms*-NiAl-P350-Sp.500h. However, its catalytic performance was unaffected. It was observed that the large metal particles were mainly composed of Ni species (Fig. 4b), without obvious P occupations (Fig. 4h). The XPS spectra showed that the chemical states of Ni, Al and P species after 500 h reaction remained unchanged compared with that of *ms*-NiAl-P350-Sp.24h (Figs. 4e-g), except for a slightly more pronounced peak being assigned to the metallic Ni (Fig. 4e). It is possible to infer that the Ni species inside the NiAl-oxide skeleton without P-doping are prone to be reduced to form large Ni nanoparticles during the reaction, which has less affection to the surface FLP structure. So, the FLP catalyst of *ms*-NiAl-P350 has excellent long-term stability under hydrotreating conditions without the need for phosphorus replenishment.

3. The combination of applied and fundamental approaches here reads as the combination of 2 separate papers, but these do not bring a high impact result. The results presented from the bulk P-Ni-Al oxide (Figure 5), which is the more scalable version of the catalyst, demonstrate that all the materials chemistry associated with the

porous material is not needed and the whole focus of the paper on that material is unnecessary - it should be focused on this more scalable material.

Response: We would like to thank the reviewer for this suggestion. As known, catalytic activity is one of the most important criteria for screening catalysts both in basic research and industrial production. In this work, the *ms* template-induced biological structure plays an important role in improving accessibility of the active sites, thereby leading to an enhanced catalytic activity. In addition, this work is expected to inspire more research on the design of porous biological structural catalysts using waste biomass. Therefore, we consider it necessary to focus our attention on the *ms*-NiAl-P350 catalyst, where we have conducted an in-depth investigation into its structure and composition before and after use, and have determined the catalytic nature of the FLP configuration.

At the end of the manuscript, we presented the bulk FLP catalyst (NiAl-P350) as an additional option for industrial applications. As the catalytic mechanism has already been demonstrated in detail using *ms*-NiAl-P350 catalyst, it is not necessary to repeat the comprehensive characterization of the bulk FLP catalyst. So, we think that combination of fundamental and applied approaches in this manuscript could yield significant impact and garner greater attention from readers in the industry, which meets the standards required for *Nature Communications*. However, to address the reviewer's concern, we have moved the Fig. 5d (catalytic data of bulk catalyst) into the Supplementary Materials section (see Supplementary Fig. 32), and added a brief discussion about the comparison between *ms*-NiAl-P350 and bulk NiAl-P350 in the revised manuscript.

*“Furthermore, we prepared the bulk NiAl-P350 catalyst (without ms-template) on a kilogram scale and shaped it into extrudates (Supplementary Fig. 32), over which a stable conversion of methyl laurate (~80%) with high selectivity towards C₁₁H₂₄ (~90%)/C₁₂H₂₆ (~4%) was achieved during 1000 h of operation at WHSV of 9.4 h⁻¹. The reaction efficiency over the bulk NiAl-P350 catalyst (36.4 mmol·g⁻¹·h⁻¹) is lower than that over the *ms*-NiAl-P350 catalyst (117.9 mmol·g⁻¹·h⁻¹), which should be attributed to its limited accessibility of active sites, given that the two catalysts possess similar chemical characteristics as determined by the HRTEM, XRD and XPS analysis*

(Supplementary Figs. 33 and 34). This comparison demonstrates that replicating the biological structure of ms to generate hierarchical pores is a highly effective approach to enhance the catalytic performance of such FLP-type catalysts. However, the bulk NiAl-P350 would be a suitable alternative for industrial applications, taking into account the expense and magnitude of catalyst production.” (Page 16)

4. From the applied catalysis perspective, the catalytic data was not presented with the appropriate level of rigor to show that the "ms" version of the catalyst provides any advantage other than surface area. That is, there are no site density measurements that would enable a turnover frequency analysis, which would show that the 2 catalyst materials are operating the same or differently?

Response: We thank the reviewer for this valuable suggestion. A turnover frequency analysis could bring more information about the intrinsic activity if the number of active sites is precisely determined. However, it is difficult to accurately measure the number of frustrated Lewis pairs (i.e. Ni(-O-Al)··P) on the surface of our present catalyst. In order to ensure the precision of catalytic data, we did not carry out the turnover frequency analysis in this study, but we have calculated the normalized reaction efficiency in the revised manuscript to distinguish the catalytic activity more clearly. It is found that the reaction efficiency over the *ms*-NiAl-P350 ($117.9 \text{ mmol} \cdot \text{g}^{-1} \cdot \text{h}^{-1}$) is higher than that over the bulk NiAl-P350 ($36.4 \text{ mmol} \cdot \text{g}^{-1} \cdot \text{h}^{-1}$). In the next work, the turnover frequency analysis will be studied in detail based on the catalyst, where the Ni(-O-Al)··P pairs are directly constructed on the surface of an Al₂O₃ carrier.

5. Thus, this combination of results would be better served as 2 separate papers: one focused in an applied catalysis journal where the preparation, characterization, and catalytic data from the non-structured, bulk material is presented. This would enable stronger arguments about the potential to replace sulfided catalysts. Then, a second paper could focus on the interesting materials chemistry using a structure directing agent that enables the production of high surface area analogs to these bulk materials. Some metric of performance must be found that differentiates these materials beyond

an increase surface area, and perhaps further structural analysis would find this.

Response: We appreciate the reviewer's valuable suggestion. It is generally reasonable to publish a first paper focusing on the bulk material and then a second one based on the materials with high surface area. However, in this manuscript, we have focused on the “*ms*” porous structured FLP catalyst with high surface area, where the preparation, characterization, and catalytic data as well as the catalytic mechanism have been well presented. In this case, we feel that it is not necessary to repeat the similar experiments and characterization on the bulk catalyst for publishing the second paper. The data of bulk catalyst in our manuscript serves as evidence to prove the industrial viability of FLP catalyst, which can draw the attention of readers from the industry and make it more suitable for publishing in the journal of *Nature Communications* with a broadly interested readership. We sincerely hope that the reviewer finds our responses satisfactory.

Response to Reviewer 3

Referee Letter:

In this submission, the authors reported a frustrated Lewis pair catalyst to realize efficient green diesel production, and investigated its charge transfer mechanism in H₂ activation. However in my opinion, the results and analysis is not sufficient to support their principle and conclusion.

Response: We appreciate the reviewer for these valuable comments and constructive suggestions, which are very helpful in improving our manuscript.

1. For the frustrated Lewis pair of Ni(-O-Al)•••P, the more positive Ni site caused by electron transfer from Ni-O to the nearby O-Al is an extremely important factor. The authors demonstrated this electron transfer by the higher white line intensity of Ni K-edge, and the existence of Ni δ^+ ($2 < \delta < 3$) in Ni 2p_{3/2} XPS spectra. However, the more important and direct evidence is lacked, which is the comparison in Ni and Al XPS spectra between pure NiO or Al₂O₃ and Al-doped NiO.

Response: We thank the reviewer for this helpful suggestion. According to the reviewer's suggestion, we have prepared the *ms*-NiO and *ms*-Al₂O₃ by a similar procedure to that of *ms*-NiAl, and analyzed their Ni and Al XPS spectra. The Ni 2p_{3/2} XPS spectrum of *ms*-NiO (Supplementary Fig. 12a and Table 4) showed two peaks at 855.9 eV and 854.0 eV, which were assigned to Ni³⁺ and Ni²⁺, respectively. In the case of *ms*-NiAl, a new peak belonging to Ni^{δ+} peak (2<δ<3) appeared at 855.7 eV, while the peaks of Ni³⁺ (856.9 eV) and Ni²⁺ (854.3 eV) both shifted to higher binding energy. In Al 2p region (Supplementary Fig. 12b), the binding energy of *ms*-NiAl (74.1 eV) shifted to a lower value compared with that of *ms*-Al₂O₃ (74.3 eV). These results illustrate that the interaction between Al³⁺ and NiO was constructed in *ms*-NiAl, wherein electrons transfer from Ni-O to the nearby O-Al. The data has been added to the Supplementary Materials section (see Supplementary Fig. 12).

Supplementary Fig. 12. (a) XPS Ni 2p spectra of *ms*-NiAl and *ms*-NiO. (b) XPS Al 2p spectra of *ms*-NiAl and *ms*-Al₂O₃.

2. In model 2, why is the crystal structure of the top Ni-Al-O layer is vastly different from that of the bottom layer? The top layer seems to be close to amorphous nature.

Response: Based on the crystal structure of NiO (111), we constructed the initial model of NiAl oxide by replacing 1/3 of the nickel atoms with Al atoms (Fig. R1). Therein, the top two layers were regarded as the crystal surface, while the bottom two layers

were used to simulate the inside crystal structure, which has a limited effect on the surface structure and catalytic reaction. Therefore, we optimized the equilibrium lattice constant of the top two layers while fixing the bottom two layers. The results indicate that the addition of Al indeed causes crystal reconstruction to a certain extent, making the top layer close to amorphous nature.

In the XRD pattern (Fig. R2), the *ms*-NiAl only shows broad diffraction peaks belonging to the NiO phase without diffraction peaks ascribed to the Al-containing phase, indicating that the Al species is amorphous or incorporated into the NiO phase or both. Based on the Scherrer's equation, the size of the NiO phase in *ms*-NiAl (7.8 nm) is much smaller than that in *ms*-NiO (52.2 nm), indicating that the Al species has a confinement effect on the growth of NiO nanoparticles during calcination. By combining XRD analysis with the DFT model, it can be inferred that the surface of *ms*-NiAl is close to amorphous in nature, due to the formation of Al³⁺-doped NiO-type Ni-Al mixed oxides.

Fig. R1 Initial and optimized model 2 simulating the structure of *ms*-NiAl.

Fig. R2 XRD spectra of *ms*-NiAl and *ms*-NiO.

3. According to P 2p XPS spectra, the peak at 133.8 eV in *ms*-NiAl-P350 is arises from P-O species, such as PO₃, PO₄, P₂O₅, etc. In the model 3, the PH group is bonding to one O atom. I don't think this P-relevant construction is accurately representative for its actual structure reflected by XPS.

Response: We thank the reviewer for this insightful comment. For the P 2p XPS spectrum of *ms*-NiAl-P350, the peak assigned to P-O species can be further deconvoluted into two peaks (Fig. R3), wherein the sub-peak at 134.1 eV is attributed to phosphate species, while the sub-peak at 133.0 eV can be assigned to the P-O bond (*ACS Energy Lett.* 2018, 3, 892-898; *ACS Catal.* 2018, 8, 8420-8429), confirming the validity of the P configuration in model 3.

Actually, we had tried several possible configurations to simulate and screen the active sites of the *ms*-NiAl-P350 catalyst, which were not suitable due to their high energy barrier in H₂ adsorption/activation. For instance, we tried to construct models in which the -PH group is bonded to another hydroxyl group (Fig. R4), however, the H₂ molecules are unable to be stably adsorbed either on the P or the adjacent O/Al/Ni atoms. In addition, DFT calculations showed that the nudged elastic band (NEB) method could not locate the transition state of the H₂ dissociation reaction at the bridge

site between Ni \cdots P (or Ni \cdots OH), indicating that these sites are not active sites for the H₂ dissociation.

In addition, we also tried to attach the -PH to the inner atoms, and the P atom eventually binds with two O atoms and one Ni atom after optimization (Fig. R5). This configuration may partially contribute to the appearance of a phosphate peak in P 2p XPS spectrum. However, the inner layer atoms are not effective active sites for catalytic reactions.

Fig. R3 Deconvolution of the XPS P 2p spectrum of *ms*-NiAl-P350.

Fig. R4 Initial and optimized configuration of H₂ adsorption on different positions in Ni(-O-Al)···P FLP.

Fig. R5 Optimized model 4 in which the P atom is inserted into the inner layer and eventually bonded to two O atoms and one Ni atom.

4. BTW, the a PH₂ group was used in model 3 of Supplementary Fig. 17 by mistake.

Response: We thank the Reviewer for pointing out this mistake. The group mentioned by the reviewer in model 3 should be a -PH group, which has been corrected in the revised version.

5. According to the elemental contents of *ms*-NiAl-P350 revealed by ICP-OES (0.44 P/Ni atomic ratio) or XPS (0.91 P/Ni atomic ratio), the P/Ni atomic ratio in model 3 differs too much. So the sufficient P doping in bulk should be considered in DFT model, not just the PH group on the surface.

Response: We agree with the reviewer that there should be multiple P atoms in the model structure of *ms*-NiAl-P350. According to the reviewer's suggestion, we added multiple -PH groups to the surface and inner layers of model 3 and optimized the structure to obtain model 5 (Fig. R6). It appears that the presence of additional P atoms had little effect on the charge of the atoms in the active center of model 3, which may be due to the poor electron transfer ability of NiAl-oxide.

In general, when establishing a structure model for a complicated material, a certain degree of simplification of the realistic system is inevitable, thereby enhancing

its intelligibility and universality. For example, in recent research (*Nat. Commun.* 2023, 14, 6808), the authors have simplified the model of a potassium-modified edge-rich MoS₂ catalyst (K/Mo mole ratio of 0.2) into a 6 × 6 × 1 tri-layer MoS₂ supercell with only one K atom. In this work, we established a model 3 that contains one P atom and investigated subsequent H₂ adsorption/activation around this atom, which is feasible to facilitate the calculation and make it easier to explain the energetics of Ni(-O-Al)···P FLP.

Fig. R6 The Bader charges of typical Ni, O, Al or P atoms in model 3 (with one atom) and model 5 (with four P atoms).

6. The analysis in electron transfer between adsorbed H₂ and FLP site by Bader charges is confused. Although the positive charge of Ni decreased (0.697 to 0.645), the electron of H₂ atom bonded to Ni also increased (-0.012 to -0.039). So the 0.05 electron transfer from H₂ to Ni is not convictive enough. The increased electron of Ni may be originated from its other coordination atoms (e.g. O), rather than H₂. That means the role of Ni acid site in FLP sites is not well established. The authors should ascertain it by more careful investigation.

Response: We regret that the electron transfer behavior during the H₂ activation was not well explained. This point has been clearly presented in the revised manuscript.

“As revealed by DFT calculations, when a H₂ molecule approaches the constructed FLP site, electron transfer occurs along the path of P→H1→H2→Ni, resulting in the electron loss of P atom (0.17 eV) and the electron gain of H1 atom (0.09 eV), H2 atom (0.03 eV) and Ni atom (0.05 eV) (Fig. 3b and Supplementary Fig. 17), thereby leading to the elongation of the H-H bond from 0.78 Å in gaseous H₂ to 1.83 Å in H₂ (TS).”
(Page 9)

In addition, to address the reviewer's concern, we also investigated the electron transfer number of the O atoms bonded with the Ni atom. The net electron gain/loss of the two O atoms is less than 0.01 eV, indicating that they do not directly participate in the H₂ activation process.

7. To demonstrate the advantage of FLP site, the catalytic abilities of contrast sample with single acid site (ms-NiAl) or basic site (P-doped NiO) also should be investigated. Likewise, the energy change profiles of H₂ activation process on catalysts with single acid/basic site are needed, to confirm the promotion in H-H scission by FLP site in comparison with single site catalysts.

Response: We appreciate the reviewer's helpful and constructive suggestion. It is indeed necessary to investigate the catalytic abilities of control samples with a single acid or basic site in order to demonstrate the advantage of FLP. However, the sample of ms-NiAl was unstable under the reaction conditions, and metallic Ni rapidly appeared and acted as a new active site for H₂ activation (Fig. 4a-iii and d and Supplementary Fig. 20a). In addition, it is difficult to synthesize the sample of P-doped NiO, because the NiO without incorporation of Al is easily converted to nickel phosphides (Ni_xP_y) during the phosphorization process.

To further confirm the necessity of FLPs in this catalytic system, we carried out a series of site-poisoning experiments. As known, pyrrole (strong Lewis acid) and pyridine (strong Lewis base) can be absorbed on the Lewis basic and acidic sites, respectively (*Nat. Commun.* 2017, 8, 15266). Therefore, the FLPs can be effectively blocked by the pyridine and pyrrole molecules. It was found that after adding 20 wt% of pyrrole/pyridine into the catalytic system, the yield of target products decreased from

16.4% to 1.2% (pyrrole) and 2.1% (pyridine) at 300 °C, and from 82.0% to 33.6% (pyrrole) and 36.6% (pyridine) at 350 °C (Supplementary Fig. 30). These observations emphasized the necessity of Ni(-O-Al)···P FLPs construction for activating H₂ in order to achieve a high activity in the present catalytic system. Corresponding changes have been made to the manuscript on page 14-15, and in the Supplementary Materials (Supplementary Fig. 30).

Supplementary Fig. 30. Influences of molecular Lewis-base or Lewis-acid on the catalytic activity of Ni(-O-Al)···P FLP for hydrotreating methyl laurate (ML) to produce C₁₁H₂₄ and C₁₂H₂₆. Reaction conditions: P_{H2}=3.0 MPa, WHSV=28.3 h⁻¹. The content of pyrrole /pyridine in the feedstock is 20 wt.%.

8. As shown in Fig 3d, the conversion of methyl laurate kept at 85-88%, with selectivity of C₁₁+C₁₂ at 91.5%+4.5% in the long-term running of *ms*-NiAl-P350 (350 °C, WHSV=28.3 h⁻¹) However in Fig 4d, the conversion (red star) is quite closed to 100%.

Response: Based on the reviewer's description, we think that the designated one is Fig. 3f (Fig. 4d shows the XRD patterns of the fresh and spent *ms*-NiAl). In Fig. 3f, the vertical axis is not “Conversion” but “Activity retention”, which is used to compare the stability of various catalysts. It is calculated using the following equation:

$$\text{Activity retention} = \frac{\text{Final conversion}}{\text{Initial conversion}} \times 100\%$$

The definition has been explained following Supplementary Table 6.

REVIEWER COMMENTS

Reviewer #1 (Remarks to the Author):

The authors have reorganized somewhat the information contained in the manuscript to avoid unnecessary duplications following the recommendation of the reviewer.

However, there is still pending a major issue. This is the comparison with the bulk catalyst. The authors claim that the the catalytic efficiency of the templated material is about three times higher than that of the bulk one, which is assigned to the limited accessibility of the latter. However, this conclusion is no so evident. The bulk catalyst presents a surface area relatively close to that of the templated ones. Moreover, the substrate conversion is also very high (around 80 %) and stable along the time on stream when using the bulk catalyst.

Therefore, to support that conclusion, which is one of the most importants in the work, the authors should compare in a clear way the physicochemical properties and the catalytic performance (in similar conditions at varying WHSV values) of both templated and bulk catalysts, indicating how the catalytic efficiency is calculated in those experiments.

Reviewer #3 (Remarks to the Author):

In the resubmission, the authors strengthened their manuscript by supplementing lots of experiments and explanations, to the key issues I concerned, e.g. the electron transfers within the FLP site and while H₂ activation, the DFT model construction. Now the the conclusions about FLP site could be supported by their experimental evidence and analysis basically. I think it could be accepted by Nature Communications at this stage.

Response to Reviewers' Comments

Response to Reviewer 1

Referee Letter:

The authors have reorganized somewhat the information contained in the manuscript to avoid unnecessary duplications following the recommendation of the reviewer.

However, there is still pending a major issue. This is the comparison with the bulk catalyst. The authors claim that the catalytic efficiency of the templated material is about three times higher than that of the bulk one, which is assigned to the limited accessibility of the latter. However, this conclusion is not so evident. The bulk catalyst presents a surface area relatively close to that of the templated ones. Moreover, the substrate conversion is also very high (around 80 %) and stable along the time on stream when using the bulk catalyst.

Therefore, to support that conclusion, which is one of the most important in the work, the authors should compare in a clear way the physicochemical properties and the catalytic performance (in similar conditions at varying WHSV values) of both templated and bulk catalysts, indicating how the catalytic efficiency is calculated in those experiments.

Response: We appreciate the reviewer for the valuable suggestions.

First, the catalytic performance of bulk NiAl-P350 was tested at various WHSV values (ranging from 2.8 h^{-1} to 28.3 h^{-1}), and was compared with that of *ms*-NiAl-P350 (Supplementary Fig. 33a). It is clear that the catalytic performance of *ms*-NiAl-P350 is always superior to that of bulk NiAl-P350 under the same reaction conditions.

Supplementary Fig. 33. (a) Conversion of methyl laurate as a function of WHSV value over *ms*-NiAl-P350 and bulk NiAl-P350. (b) The calculated reaction rates over *ms*-NiAl-P350 and bulk NiAl-P350. Reaction conditions: T=350 °C, P_{H₂}=3.0 MPa.

The catalytic reaction rates were calculated using the equation:

$$R \text{ (mmol}\cdot\text{g}^{-1}\cdot\text{h}^{-1}) = \frac{\text{WHSV} \times X_{ML}}{214.3} \times 1000$$

Where WHSV (h^{-1}) is the mass flow rate of methyl laurate divided by the mass of catalyst, X_{ML} (%) denotes the conversion of methyl laurate, 214.3 ($\text{g}\cdot\text{mol}^{-1}$) is the molar mass of methyl laurate.

To better compare the catalytic activities of the two catalysts, their reaction rates should be calculated and compared before a complete conversion and under a unified standard. When compared at identical conversion levels, the reaction rate is calculated to be $92.8 \text{ mmol}\cdot\text{g}^{-1}\cdot\text{h}^{-1}$ for *ms*-NiAl-P350 (87.9% conversion, entry 5, Supplementary Fig. 33b), which is about four times higher than that for bulk NiAl-P350 (85.8% conversion, entry 2, Supplementary Fig. 33b). If the WHSV value is maintained constant, the reaction rate over *ms*-NiAl-P350 is about two times higher than that over bulk NiAl-P350 (entry 5 and 6, Supplementary Fig. 33b).

In fact, the *ms*-NiAl-P350 and bulk NiAl-P350 were synthesized with similar procedures (except for the different shaping methods), and they possess similar chemical characteristics as determined by the HRTEM, XRD and XPS analysis (Supplementary Figs. 35 and 36). Therefore, it is reasonable to assume that the structure of the active sites in the two catalyst is similar.

Supplementary Fig. 36. Comparison of structural characterization between *ms*-NiAl-P350 and bulk NiAl-P350: (a) XRD patterns, XPS spectra in (b) Ni 2p_{3/2} region, (c) P 2p region and (d) Al 2p region.

To figure out why *ms*-NiAl-P350 works better than bulk NiAl-P350, their porous structures were further investigated. As shown in Supplementary Figs. 34a-b, the *ms*-NiAl-P350 has well-developed porous channels replicated from the *ms*-template, but there are only a few macropores visible in the bulk NiAl-P350. According to the mercury intrusion porosimetry measurement (Supplementary Fig. 34c), the total intrusion volume of *ms*-NiAl-P350 is 44.5 cm³·g⁻¹ due to the presence of abundant macropores (ranging from 30-120 μm), which is much higher than that of bulk NiAl-P350 (0.3 cm³·g⁻¹).

Supplementary Fig. 34. (a) SEM images of *ms*-NiAl-P350. (b) SEM images of bulk NiAl-P350. (c) Pore size distributions of *ms*-NiAl-P350 and bulk NiAl-P350 measured by mercury intrusion porosimetry.

It is well known that a heterogeneous catalytic reaction typically goes through the following steps: (1) external mass transfer, (2) internal mass transfer, (3) adsorption, (4) catalytic reaction, (5) desorption, (6) internal mass transfer, and (7) external mass transfer (*Ind. Eng. Chem. Res.* 2023, 62, 7769-7838). Appropriate design of porous properties can facilitate the acceleration of internal mass transfer and enhance the exposure of active sites, thereby optimizing the catalytic activity of catalysts (*Chem. Soc. Rev.*, 2017, 46, 481-558; *J. Mater. Chem. A*, 2017, 5, 8825-8846; *Science*, 2018, 359, 206-210). When powdery catalysts are molded into monoliths by industrial shaping

methods (binders or high pressure are often involved), their porosity will be significantly reduced, leading to a decline in catalytic activity (*Chem. Soc. Rev.*, 2013, 42, 6094-6112; *ACS Catal.* 2014, 4, 2409-2417; *Nano Res.* 2023, 16, 11358-11365).

Considering the differences in BET surface area of *ms*-NiAl-P350 ($103.4 \text{ m}^2 \cdot \text{g}^{-1}$) and bulk NiAl-P350 ($78.3 \text{ m}^2 \cdot \text{g}^{-1}$), together with the aforementioned differences in porosity, it can be inferred that the lower catalytic activity of bulk NiAl-P350 is attributed to its relatively lower porosity and surface area, which limit the internal mass transfer and accessibility of active sites. Corresponding changes have been made to the manuscript on page 16, and in the supplementary materials on page 36-39 (Supplementary Figs. 33-36).

Response to Reviewer 3

Referee Letter:

In the resubmission, the authors strengthened their manuscript by supplementing lots of experiments and explanations, to the key issues I concerned, e.g. the electron transfers within the FLP site and while H₂ activation, the DFT model construction. Now the the conclusions about FLP site could be supported by their experimental evidence and analysis basically. I think it could be accepted by Nature Communications at this stage.

Response: We thank the reviewer for the recognition of our work and for the time reviewing this manuscript.

REVIEWERS' COMMENTS

Reviewer #1 (Remarks to the Author):

In the last revised version the authors have addressed in a suitable way the pending issue stated by the reviewer.

They have included additional experimental data and comments that now evidence clearly that the templated catalyst presents a performance quite superior to that of the bulk one.

Therefore, I recommend its acceptance for publication.

Response to Reviewers' Comments

Response to Reviewer 1

Referee Letter:

In the last revised version the authors have addressed in a suitable way the pending issue stated by the reviewer.

They have included additional experimental data and comments that now evidence clearly that the templated catalyst presents a performance quite superior to that of the bulk one.

Therefore, I recommend its acceptance for publication.

Response: We thank the reviewer for the recognition of our work and for the time reviewing this manuscript.